# Glycation-Driven Inflammation: COVID-19 Severity in Pregnant Women and Perinatal Outcomes

**DOI:** 10.3390/nu14194037

**Published:** 2022-09-28

**Authors:** Daniela Di Martino, Mattia Cappelletti, Marta Tondo, Katia Basello, Camilla Garbin, Attilio Speciani, Enrico Ferrazzi

**Affiliations:** 1Department of Woman, Child and Neonate, Fondazione IRCCS Cà Granda Ospedale Maggiore Policlinico, 20122 Milan, Italy; 2GEK srl, 20149 Milan, Italy; 3Department of Clinical and Community Sciences, University of Milan, 20122 Milan, Italy

**Keywords:** SARS-CoV-2, pregnancy, methylglyoxal, glycated albumin, AGE–RAGE oxidative stress

## Abstract

The link between being pregnant and overweight or obese and the infectivity and virulence of the SARS CoV-2 virus is likely to be caused by SARS-CoV-2 spike protein glycosylation, which may work as a glycan shield. Methylglyoxal (MGO), an important advanced glycation end-product (AGE), and glycated albumin (GA) are the results of poor subclinical glucose metabolism and are indices of oxidative stress. Forty-one consecutive cases of SARS-CoV-2-positive pregnant patients comprising 25% pre-pregnancy overweight women and 25% obese women were recruited. The aim of our study was to compare the blood levels of MGO and GA in pregnant women with asymptomatic and symptomatic SARS-CoV-2 infection with pregnant women without SARS-CoV-2 infection with low risk and uneventful pregnancies and to evaluate the relative perinatal outcomes. The MGO and GA values of the SARS-CoV-2 cases were statistically significantly higher than those of the negative control subjects. In addition, the SARS-CoV-2-positive pregnant patients who suffered of moderate to severe COVID-19 syndrome had higher values of GA than those infected and presenting with mild symptoms or those with asymptomatic infection. Premature delivery and infants of a small size for their gestational age were overrepresented in this cohort, even in mild-asymptomatic patients for whom delivery was not indicated by the COVID-19 syndrome. Moreover, ethnic minorities were overrepresented among the severe cases. The AGE–RAGE oxidative stress axis on the placenta and multiple organs caused by MGO and GA levels, associated with the biological mechanisms of the glycation of the SARS-CoV-2 spike protein, could help to explain the infectivity and virulence of this virus in pregnant patients affected by being overweight or obese or having gestational diabetes, and the increased risk of premature delivery and/or low newborn weight.

## 1. Introduction

The COVID-19 syndrome caused by SARS-CoV-2 infection in pregnancy is significantly correlated with maternal comorbidities, among which a predominant role is played by being overweight or obese or having gestational diabetes, which are all proinflammatory conditions [1]. The link between these comorbidities and the infectivity and virulence of this virus is likely to be caused by the SARS-CoV-2 spike protein glycosylation, which may work as a glycan shield, facilitating immune evasion, as was brilliantly proved by Zhao and co-workers [2], as well as through spike–ACE2 interactions [3]. This association between COVID-19 severity and the serum concentration of the soluble receptor for advanced glycation end-products (sRAGE) has been proven in a cohort of 145 adult subjects. Indeed, the COVID-19 inflammatory process in patients with pneumonia is characterized by an increase in these membranes or in soluble RAGE, which activate a cascade that eventually recruits inflammatory cells [4,5]. This upregulation of RAGE, among other oxidative stress-related molecules, was observed in another study on adult patients with severe COVID-19 syndrome [6].

Methylglyoxal (MGO) is an important advanced glycation end-product (AGE) generated mainly by carbohydrate consumption, as well as by lipid and protein metabolism. This molecule, when above functional levels, is linked to insulin resistance and endothelial dysfunction [7]. It might also damage the beta-pancreatic cell. The AGE-RAGE axis upregulates oxidative stress and chronic inflammation and, as such, the MGO levels in pregnant SARS-CoV-2 patients could be the link to, or a facilitator of, pulmonary inflammation, severe respiratory syndrome, and placental infection [8].

An additional dysfunctional molecule of interest that could link the infection with severe syndrome is glycated albumin (GA). Albumin, when exposed to glucose or glycating substances, given its multiple glycation sites, shows a glycation speed that is approximately 4.5 times that of glycated hemoglobin [9]. For this reason, a glycated protein better represents poor glycemic control in the two-to-three weeks prior to testing and is unaffected by erythrocytes metabolism, which is accelerated in pregnancy. Because of these reasons, GA has proved to better predict fetal macrosomia and pregnancy complication in gestational diabetes than glycated hemoglobin [10]. This rapid glycation can also be of interest in investigating the relationship between major metabolic co-morbidities in pregnancy and severity of SARS-CoV-2 infection [8].

The aim of our study was to compare the blood levels of methylglyoxal and glycated albumin in pregnant women, with asymptomatic and symptomatic SARS-CoV-2 infection, and those of pregnant women without SARS-CoV-2 infection, with low risk and uneventful pregnancies, and to evaluate the relative perinatal outcomes.

## 2. Materials and Methods

### 2.1. Study Design

Since March 2020, we have conducted a national study [11] and an international multicenter study [1] to investigate the impacts of the SARS-CoV-2 infection in consecutive cohorts of pregnant patients admitted to our COVID-19 hub maternity hospital in Lombardy, Italy.

In October 2021, we amended our Co-OST study and obtained approval from the Ethical Committee of Milan Area 2 (Co-OST, n° 295_2021). The amendment included the dosage of MGO and GA at admission in SARS-CoV-2 infected pregnant patients and the recruitment of a control group. Subjects were enrolled at their admission to the COVID-19 Obstetric Wards. SARS-CoV-2 infection was diagnosed by PCR RNA analysis of nasopharyngeal swabs. The inclusion criteria were a maternal age of ≥18 years and the ability to sign the informed consent. The present study included patients recruited consecutively from the approval of the protocol amendment. The control subjects were SARS-CoV-2-negative, low-risk singleton pregnancies recruited in a 1:1 ratio in accordance with the same inclusion criteria as regards maternal age and gestational age at the time of the consultation at the obstetrics outpatient clinics, who eventually delivered with no complications for the mother or the fetus.

The exclusion criteria for all patients were: multiple pregnancy, fetal malformation, and other maternal infections during pregnancy.

The severity of the infection was classified according to the clinical COVID-19 syndrome, as follows: pregnant women with asymptomatic or mildly symptomatic SARS-CoV-2 infection (such as anosmia, ageusia, cold, or sore throat) (Group 1) and pregnant women with moderate or severe SARS-CoV-2 infection symptoms who needed oxygen supply (both non-invasive and mechanical ventilation) (Group 2) [12].

All subjects underwent routine fetal clinical assessments, including ultrasonography and Doppler velocimetry examinations of the fetal and placental arteries. Clinical data, delivery, and neonatal outcomes were recorded.

### 2.2. Methylglyoxal and Glycated Albumin Levels Measurement

Methylglyoxal and glycated albumin were assayed on maternal venous blood that was sampled at recruitment as follows: SARS-CoV-2-negative patients were sampled at the time of recruitment, while SARS-CoV-2-positive patients were sampled during admission, though always before starting low molecular weight heparin, if needed, to treat the COVID-19 syndrome. The sampled blood was collected using Copan 552C swabs (150 mcl of dried blood). The swabs were bar-code labelled, signing as positive or negative for COVID-19 infection for the different analysis in the laboratory.

Methylglyoxal was analyzed in the GEK srl laboratory using an OxiSelect™ Methylglyoxal Competitive ELISA Kit (LLD 0 µg/mL, Catalog No. STA-811, Cell Biolabs, San Diego, CA, USA), which is an immune-enzymatic system developed to quantify methylglyoxal-hydro-imidazoline (MG-H1) products.

Glycated albumin was analyzed in the GEK srl laboratory using a Human Glycated Albumin ELISA Kit (LLD 19.53 pmol/mL, sensibility < 11.719 pmol/mL, Catalog No. abx252493, Abbexa Ltd., Cambridge Science Park, Cambridge, UK) and a Human Albumin Immunoperoxidase Assay for the Determination of Albumin kit (LLD 0 ng/mL, Human Samples, Catalog No. E-80AL, Immunology Consultants Laboratory, Portland, OR, USA). The results were expressed as percentages of total serum albumin using the formula GA% = GA (μmol/mL)/total albumin (μmol/mL) × 100.

### 2.3. Statistical Analysis

Non-parametric tests were used for the univariate analysis. Given the non-normal distribution, we used the Mann–Whitney U test and ANOVA of the Kruskall–Wallis test for the scalar variables, with a Dunn’s test for multiple comparisons, and Fisher’s exact test and the Marascuilo procedure for the categorical variables. The data were expressed by medians and interquartile ranges (IQRs) for the continuous variables and with absolute and relative frequencies for the categorical variables, respectively. The original scatterplot of measured concentrations was superimposed with a box and whiskers plot for direct visualization of the data and a non-parametric statistical comparison.

No sample size was calculated for this pilot investigation, but the cases and controls were collected during the first and second wave, characterized by a specific viral variant that was more virulent than the ones that followed. However, a purposeful sampling was obtained during the study period that considered, as the first outcome, the comparison of the blood levels of methylglyoxal and glycated albumin between the pregnant women with asymptomatic and symptomatic SARS-CoV-2 infection and the pregnant women without SARS-CoV-2 infection.

Statistical analysis of the data was performed using GraphPad Prism 9 for macOS (version 9.2.0 (283), 15 July 2021). A *p*-value of <0.05 was used as the limit of statistical significance.

## 3. Results

Forty-three consecutive SARS-CoV-2-positive pregnant women and 40 controls of SARS-CoV-2-negative pregnant women were analyzed. Fasting glycated albumin percentages and Methylglyoxal levels were obtained from all the participants. Two COVID-positive women were excluded due to post-collection mismanagement of their sample vials. The final analysis included 41 cases and 40 low-risk control pregnancies recruited in the same period that ended in uneventful outcomes.

Of the 41 pregnant patients with positive SARS-CoV-2 test results during the study time, 29 (71%) were asymptomatic or had mild illness (Group 1) and 12 (29%) had severe illness (Group 2).

At admission, in Group 1, 13 pregnant women (45%) had mild symptoms. Overall, the most common patient-reported symptoms were cough (77%), dyspnea (38%), myalgias (31%), and fever (31%). Among these patients, six (46%) had pulmonary disease on chest radiologic imaging, but none of these patients needed additional therapy or oxygen supply. Every patient received prophylactic anticoagulation. Five patients (17%) had lymphocytopenia and three (10%) developed cholestasis. The results of the other blood tests were normal. The level of SpO_2_ (%) was always above 98%.

At admission, in the 12 patients of Group 2, the most common patient-reported symptoms were cough (67%), dyspnea (75%), fever (75%), and myalgias (58%). This group of women included patients who had oxygen saturation levels of <94% or who needed oxygen supply during hospitalization. All of these patients were diagnosed with pulmonary disease on chest radiologic imaging, and nine (75%) were in need of oxygen supply after admission. Two patients (17%) were admitted to the ICU before delivery. Ten women (83%) needed additional therapy with corticosteroids because of pulmonary disease. Nine patients (75%) had lymphocytopenia, and the lowest level of SpO_2_ (%) observed was 87%.

Table 1 reports the demographic and perinatal data of the two groups and the control subjects. There was a significant difference between the two groups regarding pre-pregnancy BMI versus the control subjects. The patients with SARS-CoV-2 infection presented with higher pre-pregnancy BMIs, and the group 2 patients presented the highest values of pre-pregnancy BMIs, with 67% being overweight/obese. We observed a significantly higher percentage of ethnic minorities in groups 1 and 2 versus the control subjects.

Pre-pregnancy diabetes was observed in two patients in group 1. Gestational diabetes and hypertensive pregnancy disorders were diagnosed in nine and two patients and in four and one patients in groups 1 and 2, respectively.

In group 1, premature delivery (34%) and small-for-gestational-age fetuses (27%) were higher than expected in a normal unselected population.

Instead, there was no significant difference in the of number of fetuses with macrosomia in the groups affected by infection versus the controls.

Figure 1 shows the scatterplot, the medians, and interquartile ranges of the values of GA for the two SARS-CoV-2 groups (together) and the control subjects. The level of GA was significantly higher in the group with SARS-CoV-2 infection. Figure 2 shows the scatterplot, the medians, and interquartile ranges of the values of MGO for the two SARS-CoV-2 groups (together) and the control subjects. The level of MGO was significantly increased in the group with SARS-CoV-2 infection.

Figure 3 shows the scatterplot and comparison of the GA levels (medians and interquartile ranges) of SARS-CoV-2 groups 1 and 2 and the SARS-CoV-2-negative controls with multiple post hoc comparisons. The group 2 patients showed the highest values for GA compared to both the group 1 patients and the control group.

Figure 4 shows the comparison of the MGO levels between the SARS-CoV-2-negative controls and the SARS-CoV-2 group 1 and group 2 subjects, with multiple post hoc comparisons. Only the group with asymptomatic SARS-CoV-2 infection presented significantly higher values when compared to the control group. The Kruskal–Wallis ANOVA between the three groups showed a strongly significant difference both for the GA (*p* = 0.0001) and MGO (*p* = 0.013) levels.

## 4. Discussion

The main finding of this study was that pregnant women affected by SARS-CoV-2 infection showed higher values of glycated albumin and methylglyoxal than the SARS-CoV-2-negative control patients. Glycated albumin levels were significantly higher in the SARS-CoV-2-positive patients affected by severe COVID-19 syndrome. The association between being pregnant and overweight/obese and the infectivity and virulence of the SARS-CoV-2 virus has been widely reported [1]. Both being obese and overweight during pregnancy have adverse consequences for the mother and child. The link between these conditions of systemic low-grade inflammation and COVID-19 syndrome is likely to the result of both the pro-inflammatory role of MGO in the AGE-RAGE axis [3,4] and the glycosylation SARS-CoV-2 spike protein that may work as a glycan shield. In fact, both MGO, an important advanced glycation end-product, and GA are the results of poor subclinical glucose metabolism, and both are indices of oxidative stress. When pro-inflammatory factors associated with the fat tissues of obese and diabetic patients are added to the AGE-RAGE axis, and when glycation enhances the virulence of the virus, a perfect storm could be pending for these patients.

Moreover, we observed a significantly higher percentage of ethnic minorities in the groups affected by SARS-CoV-2 infection versus the control subjects which was twice the prevalence observed for the general non-selected population of the metropolitan area of Milan.

The observation that glycated albumin was associated with the likelihood of infection and the severity of COVID-19 syndrome is in agreement with both the role of glucose in masking the S spike epitopes and in modulating spike–ACE2 interactions [2,3]. MGO is a strong inducer of AGE formation and RAGEs [3,4], and it has been linked to insulin resistance, vascular dysfunction, and neuropathies [13]. It is also of interest that MGO-induced RAGEs play a role in platelet activation [14], which may lead to additional SARS-CoV-2 infection complications. These metabolic processes are also expressed in control pregnant subjects since pregnancy is characterized by para-physiological dyslipidemia and insulin resistance [7,15,16]. This could explain why, even in otherwise clinically normal pregnancies, we observed a wide range of GA and MGO values. We could also speculate that these direct indices of glycation could identify pregnant women at risk of infection in whom an mRNA vaccine should be proactively recommended based on biological evidence, rather than on the general principle of prevention based on epidemiological evidence.

Our findings underline the role of glycation in overweight, obese, and gestationally diabetic pregnant women as a direct biological enhancer of the infectivity and virulence of SARS-CoV-2 in pregnant women, explaining the role of these comorbidities in the clinical burden of this infection in pregnant women [17,18]. According to Eskenazi B, Rauch S, Iurlaro E, et al., the adjusted relative risk of being COVID-19-positive in overweight and obese pregnant women was 1.25, and this figure jumped to 2.04 in overweight and obese pregnant patients with gestational diabetes. Our findings provide a significant clue to this biological link, as observed by Zaho et al. [2], and this was indirectly observed in infected adults in the studies by Hoffman [3], Chavakis [4], and Saputra [5]. We could also interpret the burden of prematurity and late, small-for-gestational-age newborns in mild-asymptomatic infected mothers in whom delivery was not indicated by the syndrome, but rather, as a consequence of the pro-inflammatory glycation end-products on placenta-accelerated aging [8,10,18].

### Strengths and Limitations

These results were obtained at single center—the largest COVID-19 maternity hub in Lombardy—which might have concentrated severe cases. This could explain the fact that 12 out of the 41 cases were classified as moderate or severe, a ratio that is higher than expected in pregnancy in high income countries. Indeed, among the SARS-CoV-2-infected pregnant patients we observed, there was a significantly higher prevalence of women from minor ethnicities that are usually exposed to manual work, crowded housing, and inadequate western diets. However, the biological findings should not be altered by the relative prevalence of the moderate and severe cases. The control group was matched for maternal age and gestational age at sampling for MGO and GA, and it was not recruited as the next SARS-CoV-2 pregnancy after an index case. This offered us the chance to include MGO and GA values in uneventful pregnancies. The limitation of the clinical design of the study is that we could not compare the GA and MGO values in a control group matched for obesity and the prevalence of diabetes. However, the interpretation of the data that we are supporting here is greatly supported by the large epidemiological studies on COVID-29 syndrome that found a highly significant association of obesity and diabetes with the severity of SARS-CoV-2 infection [1,2,3,4,5,6,7,8,9,10,11,12,13,14,15,16].

Placental histology was not performed. However, we know that placental lesions in SARS-CoV-2-positive women are indicative of maternal vascular inflammation resulting in abnormalities in oxygenation within the intervillous space rather than direct viral infection [8].

## 5. Conclusions

The AGE–RAGE cascade of oxidative stress caused by MGO and GA values associated with the biological mechanisms of the glycation of SARS-CoV-2 spikes could help to explain the increased risk of infectivity and virulence of this virus in pregnant patients affected by being overweight or obesity or having gestational diabetes, as reported by large, multi-centre epidemiological studies.

## Figures and Tables

**Figure 1 nutrients-14-04037-f001:**
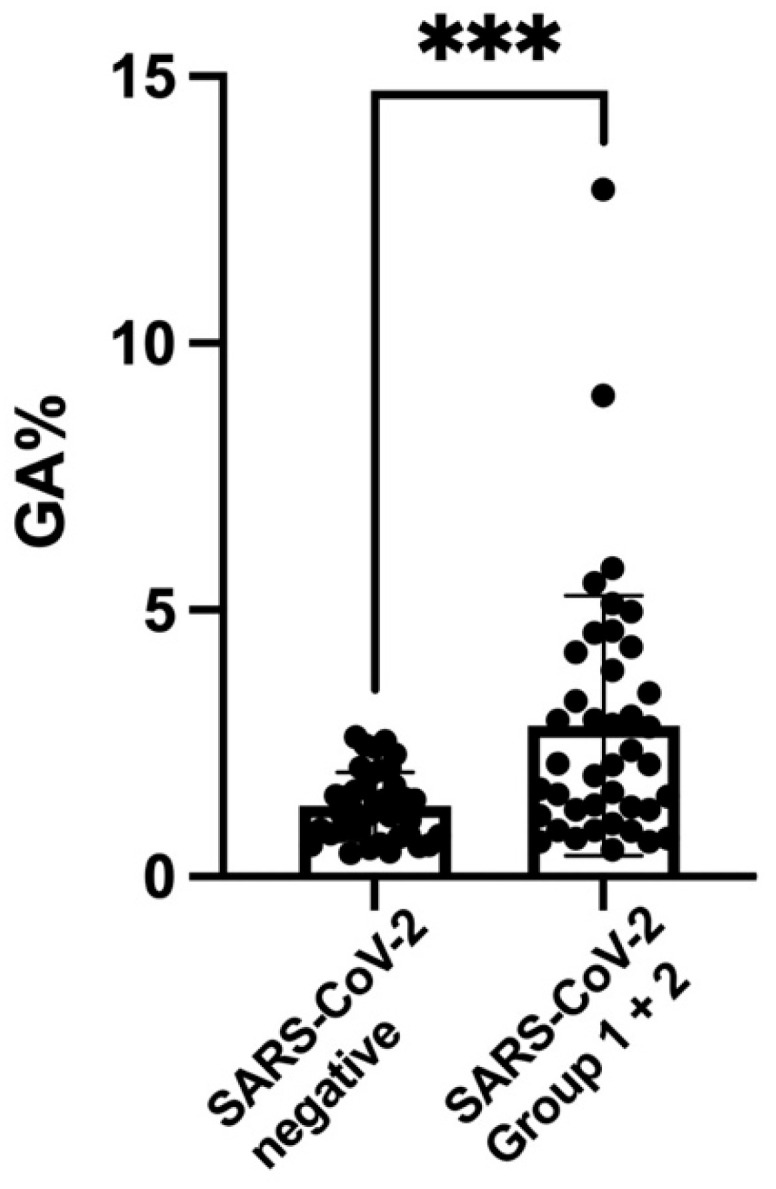
**Scatterplot of** glycated albumin (GA) **percentages (medians and interquartile ranges) of SARS-CoV-2 groups 1 and 2 (2.1; IQR 1.2–4.04) versus the SARS-CoV-2-negative controls** (1.31; IQR 0.79–1.7). *** *p*-value < 0.001.

**Figure 2 nutrients-14-04037-f002:**
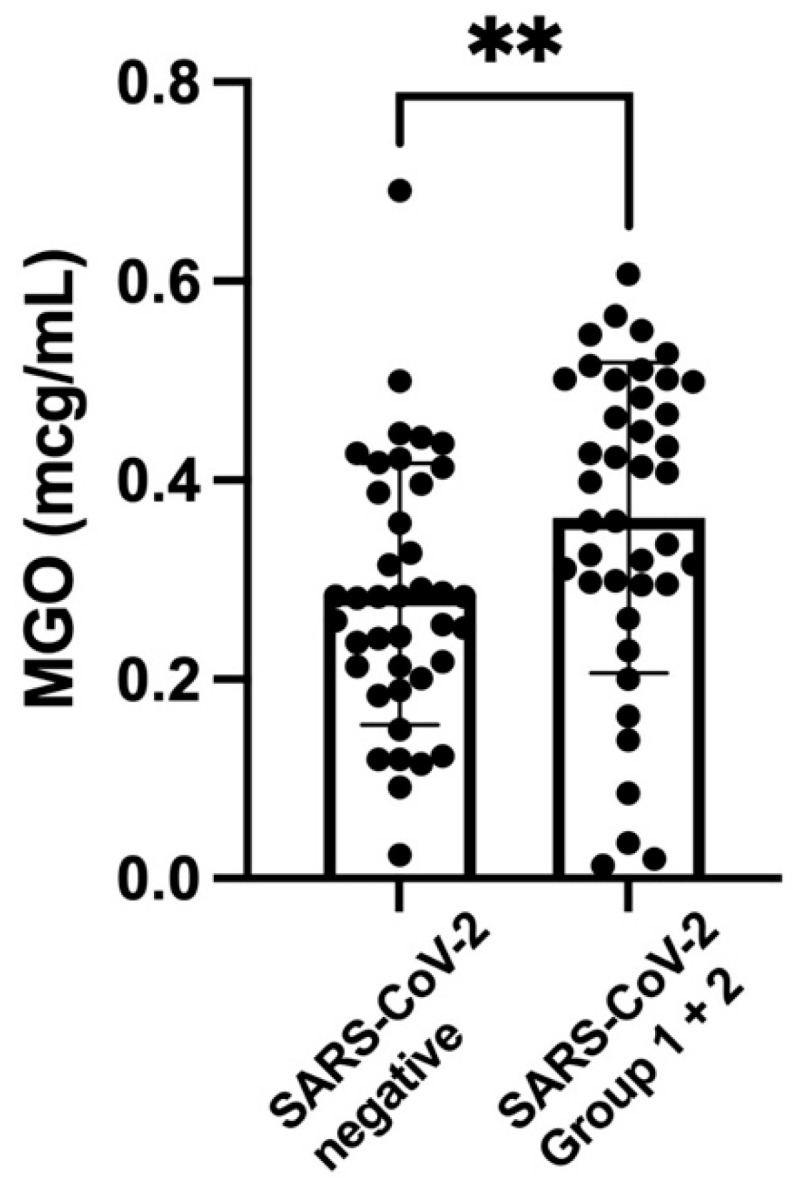
**Scatterplot of** methylglyoxal levels **(mcg/mL) (medians and interquartile ranges) of SARS-CoV-2 groups 1 and 2** (0.39; IQR 0.29–0.50) **versus the SARS-CoV-2-negative controls** (0.28; IQR 0.20–0.39). ** *p*-value < 0.003.

**Figure 3 nutrients-14-04037-f003:**
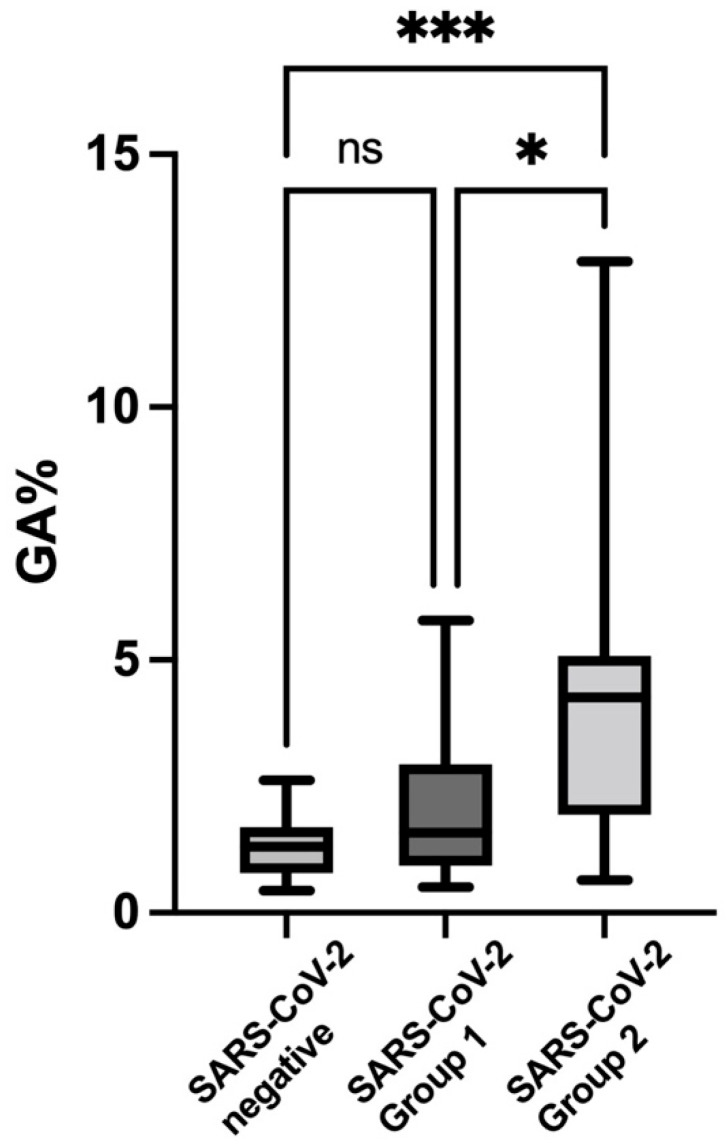
**Scatterplot of** glycated albumin **(GA) percentages (box and whiskers plot) of the SARS-CoV-2 group 1** (1.58; IQR 0.93–2.94) **and group 2 subjects** (4.26; IQR 1.94–5.08) **and the SARS-CoV-2-negative controls (see**
Figure 1
**for values), with multiple post hoc comparisons. *** *p* value < 0.001, * *p* value < 0.05**. ns: non significant.

**Figure 4 nutrients-14-04037-f004:**
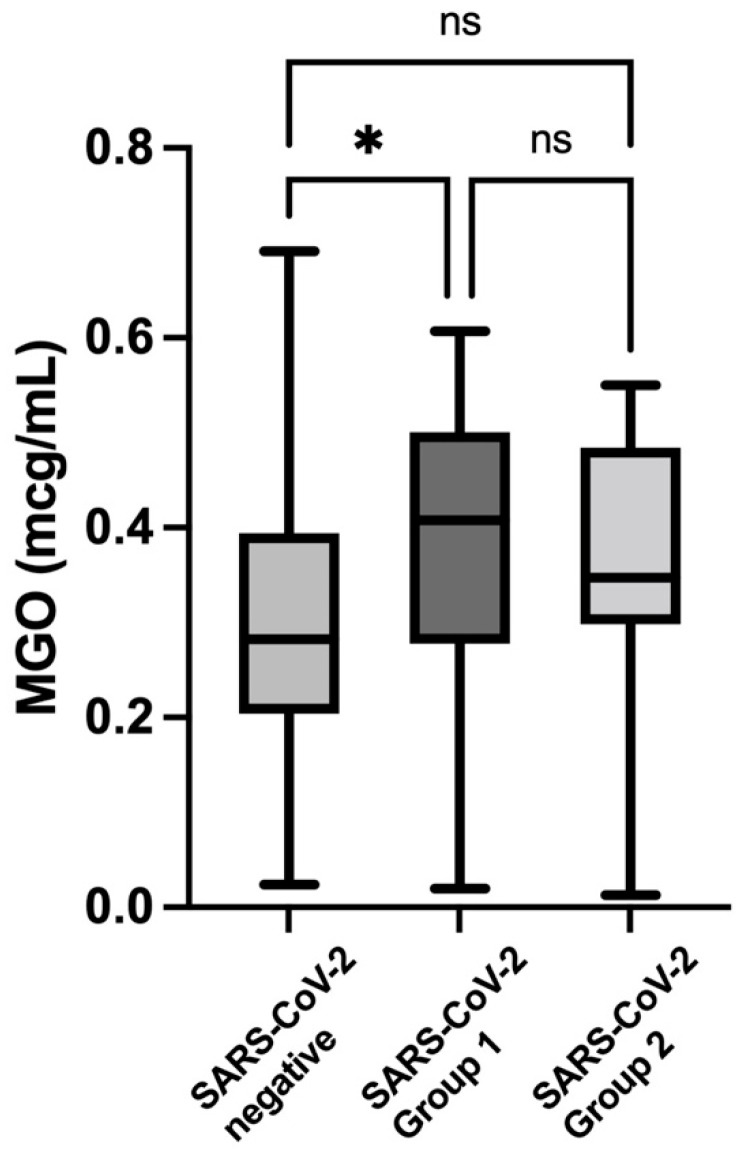
**Comparison of the methilglyoxal (MGO) levels (box and whiskers plot) between the SARS-CoV-2 group 1** (0.41; IQR 0.28–0,50) **and group 2** (0.35; IQR 0.29–0.48) **subjects and the control subjects (for values, see**
Figure 2**), with multiple post hoc comparisons. * *p* value < 0.05**. ns: non significant.

**Table 1 nutrients-14-04037-t001:** Demographic and clinical data of the three groups. The values are expressed as medians and interquartile ranges, or as percentages of the total patients in each group and the absolute number of cases, as appropriate.

Variable	SARS-CoV-2- Negative (n = 40)	SARS-CoV-2 Group 1 (n = 29)	SARS-CoV-2 Group 2 (n = 12)	*p*-Value	Post Hoc Test
Age (years)	33 (31–35)	35 (26–40)	33 (27–38)	0.7	
Pre-pregnancy BMI (kg/m^2^)	21 (19–23)	26 (22–30)	28.5 (24–31)	< 0.001	☨ ⌘
Overweight/obese (BMI ≥ 25)	6 (15%)	16 (55%)	8 (67%)	< 0.001	☨ ⌘
Multiparous (%)	6 (15%)	14 (48%)	8 (66%)	0.004	☨ ⌘
Caucasian ethnicity (%)	38 (95%)	19 (65%)	6 (50%)	0.001	☨ ⌘
Pre-pregnancy disease (%)	-	10 (34%)	2 (16%)	< 0.001	☨ ⌘
Other pregnancy complications (%)	-	21 (72%)	4 (33%)	< 0.001	☨ ⌘ ⌖
Gestational age at recruitment (weeks)	35 (34–36)	33 (29–36,5)	28,5 (24–31)	0.002	⌘ ⌖
Gestational age at delivery (weeks)	39 (39–41)	38 (35–39)	39 (37–40)	0.001	☨
Preterm delivery (%)	2 (5%)	10 (34%)	2 (18%)	0.006	☨
Neonatal weight (g)	3382 (3202–3800)	2780 (2210–3462)	3030 (2580–3600)	0.001	☨ ⌘
Newborn weight < 10° centile (%)	3 (7.5%)	8 (27.5%)	1 (9%)	0.05	☨
Macrosomia (neonatal weight > 4000 g)	3 (7.5%)	4 (14%)	-	0.3	
NICU * (%)	-	6 (21%)	2 (18%)	0.01	☨ ⌘

Post hoc tests were used to calculate the intergroup comparisons: ☨ *p*-value < 0.05 for the COVID-19-negative group vs. the COVID-19-positive group 1, ⌘ *p*-value < 0.05 for the COVID-19-negative group vs. the COVID-19-positive group 2, and ⌖ *p*-value < 0.05 for the COVID-19 group 1 vs. the COVID-19 group 2. * NICU: neonatal intensive care unit.

## Data Availability

The data presented in this study are available on request from the corresponding author. The data are not publicly available due to privacy.

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
