# Peer review of "Glycation-Driven Inflammation: COVID-19 Severity in Pregnant Women and Perinatal Outcomes"

_nutrients, 2022, doi:10.3390/nu14194037_

Round 1
Reviewer 1 Report (Previous Reviewer 1)
Thank you for addressing the previously highlighted comments.
The manuscript has been enhanced and is well presented. The results’ section and discussion have improved tremendously and have a better flow of ideas.
Minor comments should be taken into consideration:
- Typo: correct “simple size” to “sample size”
- In results section: "This is in line with pre-pregnancy [...] obese people. " Leave this out for the discussion section.
- In results section: "...that was twice the prevalence [...] area of Milan." Leave out for the discussion section.
- Figures: add IQR abbreviation definition in the methods, where it's first mentioned.
Author Response
Dear Reviewer thank you very much for your comments:
- Typo: correct “simple size” to “sample size”.
We corrected the typing error ( Line 127)
- In results section: "This is in line with pre-pregnancy [...] obese people. " Leave this out for the discussion section.
We removed the sentence "This is in line with pre-pregnancy [...] obese people. " from the results section as suggested ( Lines 224-226.)
- In results section: "...that was twice the prevalence [...] area of Milan." Leave out for the discussion section.
We removed the sentence "...that was twice the prevalence [...] area of Milan." from the results section. We moved it in the discussion section ( Lines 234-237)
- Figures: add IQR abbreviation definition in the methods, where it's first mentioned.
We added the IQR abbreviation in the methods ( Line 123).
Reviewer 2 Report (New Reviewer)
Well written and a good effort by the authors.
I was interested to know that the Authors kept the cut-off value of obese BMI > 25 because in the USA and in the UK now, the average BMI included in consultant-led care is BMI >35
Secondly, as the Authors mentioned, reference 10 which shows that GA levels are more associated with cesarean section and macrosomia.
In this study small gestation age was more obvious as compared to Macrosomia.
Question: Can this observation be due to very little difference in BMI cut-off, i.e.>25 defining obese? In women with a BMI of 35 or more, macrosomia may have been more obvious (unlike in this study where BMI cut-off defining obese is >25) and more possible to note and report.
If authors can comment, then it would be great
Author Response
Please see the attachment.

This manuscript is a resubmission of an earlier submission. The following is a list of the peer review reports and author responses from that submission.
Round 1
Reviewer 1 Report
The findings are interesting and very relevant. The findings are valuable and add knowledge to the relationship between gestational outcomes, maternal comorbidities, and infections.
Some comments should be taken into consideration to improve the manuscript as follows:
- Introduction: well written but the link AGE-RAGE can be further elaborated.
- Methods: very good study design. No mention of sample size calculation. You can also mention that it was a purposeful sampling.
- Results: Table 2 and 3 are not described in the results’ section and it is not very clear that these are supplementary material (especially that the results are also mentioned in the discussion).
- Figures are well prepared but need to be described more clearly in the results’ section. Footnotes are missing in figures for * and abbreviations.
- Findings should be mainly described in full in the result's section and less in the discussion. The "Main findings" section is not necessarily needed in the discussion.
- Line 234: Maybe conduct more analysis to examine the relationship or role played by overweight and obesity?
- Discussion: Elaborate more on the AGE-RAGE axis.
- Conclusion is one very long sentence that gets confusing. It should be re-written and recommendations could be added.
Author Response
Dear Reviewer thank you very much for your revision and suggestions. We answered your comments point by point to improve the manuscript.
- Introduction: well written but the link AGE-RAGE can be further elaborated.
A: We elaborated the link AGE-RAGE in the discussion, leaving the introduction concise and effective
- Methods: very good study design. No mention of sample size calculation. You can also mention that it was a purposeful sampling.
A: We explicated in the text that our recruitments were during the first and second pandemic wave of SARS-COV-2 infection characterized by a specific type of virus with similar virulence (Line 126-132).
- Results: Table 2 and 3 are not described in the results’ section and it is not very clear that these are supplementary material (especially that the results are also mentioned in the discussion).
A: We described Table 2 and 3 in the text as suggested (line 188-193).
- Figures are well prepared but need to be described more clearly in the results’ section. Footnotes are missing in figures for * and abbreviations.
A: We described all figures more accurately in the results section and added the footnotes and abbreviations in the figures legend (line 194-209, 219-231).
- Findings should be mainly described in full in the result's section and less in the discussion. The "Main findings" section is not necessarily needed in the discussion.
A: We moved the main findings from discussion section to result section as suggested (line 232-247).
- Line 234: Maybe conduct more analysis to examine the relationship or role played by overweight and obesity?
A: we added a link between MGO a GA levels in SARS-COV-2 infection and obesity (line 254-258).
- Discussion: Elaborate more on the AGE-RAGE axis.
A: we elaborated and added two references for the AGE-RAGE axis (line 263-266).
- Conclusion is one very long sentence that gets confusing. It should be re-written and recommendations could be added.
A: we have divided the long sentence into two sentences as suggested.
Reviewer 2 Report
The authors studied how SARS-CoV-2 infectious affect pregnancy focusing on MGO and GA, which they believe the affect health of neonatal. However, the control group have quite different background to those with SARS-CoV-2 infected. For instance, pre-pregnancy BMI, overweight status, ethnicity and comorbidities. With poorly set control, it is difficult to elucidate any convincing results.
Author Response
Dear Reviewer,
thank you very much for your revision, we will try to answer to your comments: the main outcome of our study is not the relationship between glycation and the health of the newborn, but it is the relationship between glycation in Sars-Cov-2 infected and non-infected patients.
We modified the title to focus on the main findings, as such sequencing the maternal outcome in the first position.
For our purpose we designed a 1:1 case-control study between cases and controls (41 versus 40 patients) during the first two waves of the Sars-Cov-2 pandemic periods, characterized by similar virulence.
The main finding of the study were the higher value of MGO and GA, indicating inflammatory status, in SARS Cov-2 patients in the subgroup with severe symptoms.
We also hope that the changes we made as suggested by one of the reviewers improved the paper and meets the general comments you made to our paper.
With our best regards